# Out-of-Distribution Detection & Applications With Ablated Learned Temperature Energy

**Will LeVine**[*]  **Benjamin Pikus**    **Jacob Phillips**    **Berk Norman**    **Fernando Amat**
Microsoft          Advex AI         Andreessen Horowitz       Anduril          Google

**Sean Hendryx**
Scale AI

## Abstract

As deep neural networks become adopted in high-stakes domains, it is crucial to identify when inference inputs are Out-of-Distribution (OOD) so that users can be alerted of likely drops in performance and calibration (Ovadia et al., 2019) despite high confidence (Nguyen et al., 2015) - ultimately to know when networks' decisions (and their uncertainty in those decisions) should be trusted. In this paper we introduce Ablated Learned Temperature Energy (or `AbeT` for short), an OOD detection method which lowers the False Positive Rate at 95% True Positive Rate (FPR@95) by $43.43\%$ in classification compared to state of the art without training networks in multiple stages or requiring hyperparameters or test-time backward passes. We additionally provide empirical insights as to why our model learns to distinguish between In-Distribution (ID) and OOD samples while only being explicitly trained on ID samples via exposure to misclassified ID examples at training time. Lastly, we show the efficacy of our method in identifying predicted bounding boxes and pixels corresponding to OOD objects in object detection and semantic segmentation, respectively - with an AUROC increase of $5.15\%$ in object detection and both a decrease in FPR@95 of $41.48\%$ and an increase in AUPRC of $34.20\%$ in semantic segmentation compared to previous state of the art. [2]

## 1   Introduction

In recent years, machine learning models have shown impressive performance on fixed distributions. However, in cases where inference examples are far from the training set, not only does model performance drop, nearly all known uncertainty estimates also become miscalibrated - i.e. unreliable (Ovadia et al., 2019) - although there are exceptions (Rajendran & LeVine, 2019). Without OOD detection, users can therefore be fooled into false trust in model predictions due to high confidence on OOD inputs (Nguyen et al., 2015).

Aimed at OOD detection, existing methods have explored (among many other methods) modifying models via a learned temperature which is dynamic depending on input (Hsu et al., 2020) and an inference-time post-processing energy score (Liu et al., 2020). In this paper, we combine these methods and introduce an ablation, leading to our method deemed "`AbeT`." Due to these contributions, we demonstrate the efficacy of `AbeT` over existing OOD methods. We establish state of the art

---

[*]Corresponding author email: levinewill@icloud.com

[2]We make our code publicly available at https://github.com/anonymousoodauthor/abet, with our method requiring only a single line change to the architectures of classifiers, object detectors, and segmentation models prior to training.

Workshop on Bayesian Decision-making and Uncertainty, 38th Conference on Neural Information Processing Systems (NeurIPS 2024).

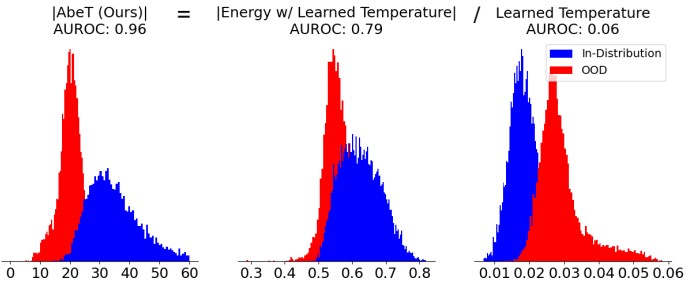

Figure 1: Histograms showing the separability between OOD scores on OOD inputs (red) and ID inputs (blue) for different methods. The goal is to make these red and blue distributions as separable as possible, with scores on OOD inputs (red) close to 0 and scores on ID inputs (blue) of high-magnitude (away from 0). (**Center**) Our first contribution is replacing the Scalar Temperature in the Energy Score (Liu et al., 2020) with a Learned Temperature (Hsu et al., 2020). This infusion leads to Equation 1, with the Learned Temperature showing up in the Exponential Divisor Temperature (overlined in Equation 1) and Forefront Temperature Constant (underlined in Equation 1) forms. (**Right**) The Forefront Temperature Constant contradicts the desired property of scores being close to 0 for OOD points (red) and of high magnitude for ID points (blue). (**Left**) Therefore, our second contribution is to ablate this Forefront Temperature Constant, leading to our final Ablated Learned Temperature Energy (AbeT) score. This ablation increases the separability of the OOD scores vs. ID scores, as can be seen visually and numerically (in terms of AUROC) comparing the center and left plots - where the only difference is this ablation of the Forefront Temperature Constant. Higher AUROC means more separability.

performance in classification, object detection, and semantic segmentation on a suite of common OOD benchmarks spanning a variety of scales and resolutions. We also perform extensive visual and empirical investigations to understand our algorithm.

## 2  Preliminaries

Let $X$ and $Y$ be the input and response random variables with realizations $x \in \mathbb{R}^D$ and $y \in \{1, 2, ..., C-1, C\}$, respectively. Typically, we'd like to make inferences about $Y$ given $X$ using a learned model $\hat{f} : \mathbb{R}^D \to \mathbb{R}^C$. In practice, a learner only has access to a limited amount of training examples in this data-set $D_{in}^{train} = \{(x_i, y_i)\}_{i=1}^N$.

### 2.1  Problem Statement

We define $D_{in}^{test}$ identical to $D_{in}^{train}$ but unseen at training time. And we define $D_{out}^{test}$ as any dataset that has non-overlapping output classes with those of $D_{in}^{train}$, as is standard in OOD detection evaluations (Huang et al., 2021; Hsu et al., 2020; Liu et al., 2020; Djurisic et al., 2022; Sun et al., 2021; Hendrycks & Gimpel, 2016; Liang et al., 2017; Sun et al., 2022; Katz-Samuels et al., 2022). The goal of Out-of-Distribution Detection is to define a score $S$ such that $S(x_{out})$ and $S(x_{in})$ are far from each other $\forall \ x_{out} \in D_{out}^{test}, x_{in} \in D_{in}^{test}$.

### 2.2  Standard Classification Model Optimization

In OOD detection, $\hat{f}$ serves a dual purpose: its outputs are optimized to classify among outputs $\{1, 2, ..., C-1, C\}$ and functions of the network are used as inputs to $S$. Though we use $\hat{f}$ for both purposes, we aim for AbeT to neither have OOD data at training time nor significantly modify training to account for the ability to detect OOD data. Thus, we train our classification models in a standard way: to minimize cross-entropy loss $\mathcal{L} = \sum_{i=1}^N -\log \hat{f}_{y_i}(x_i; \theta)$, where $(x_i, y_i) \in D_{in}^{train}$. Networks can be optimized towards other loss functions, but we did not test our method in conjunction with any other loss functions.

## 2.3 Model Output

To estimate $\hat{f}_{y_i}(x_i; \theta)$, models typically use logit functions per class which calculate the activation of class $c$ on input $x_i$ as $L_c(x_i; \theta) = \hat{g}_c(x_i; \theta)$. Let $w$ and $b$ represent the weights and biases of the final layer of a network (mapping penultimate space to output space) respectively and $f^p(x_i)$ represent the penultimate representation of the network on input $x_i$. Similar to Hsu et al. (2020), we found that a cosine-similarity-based logit function, $\hat{g}_c(x_i; \theta) = \dfrac{w_c^T f^p(x_i)}{||w_c^T|| ||f^p(x_i)||}$, was the most effective in our OOD evaluation paradigm. For more details, see Appendix Section A.3.

We now discuss tempering the logit function $L$: often employed to increase calibration, Temperature Scaling (Guo et al., 2017) geometrically decreases the logit function $L$ by a single scalar $T_{\text{scalar}}$. That is, $\hat{f}$ has a logit function that employs a scalar temperature as $L_c(x_i; \theta, T_{\text{scalar}}) = \hat{g}_c(x_i; \theta)/T_{\text{scalar}}$. Introduced in Hsu et al. (2020), a learned temperature $T_{\text{learned}} : \mathcal{X} \to (0,1)$ is a temperature that depends on input $x_i$. That is, $\hat{f}$ has a logit function that employs a learned temperature $L_c(x_i; \theta, T_{\text{learned}}) = \hat{g}_c(x_i; \theta)/T_{\text{learned}}(x_i)$. The softmax of this tempered logit serves as the final model prediction: $\hat{f}_{y_i}(x_i; \theta) = \dfrac{\exp\left(\hat{g}_{y_i}(x_i; \theta)/T_{\text{learned}}(x_i)\right)}{\sum_{c=1}^{C} \exp\left(\hat{g}_c(x_i; \theta)/T_{\text{learned}}(x_i)\right)}$. This is input to the loss $\mathcal{L}$ during training. In this formulation, $\hat{g}$ and $T_{\text{learned}}$ serve disjoint purposes: $\hat{g}$ reduces loss by selecting $\hat{g}_{y_i}(x_i; \theta)$ as highest among $\{\hat{g}_c(x_i; \theta)\}_{c=1}^{C}$; and $T_{\text{learned}}(x_i)$ reduces loss by modifying the confidence (but not changing the classification) such that the confidence is high or low when the model is correct or incorrect, respectively.

We provide a visual architecture of a forward pass with a learned temperature in Appendix Figure 3.

For more information on the details of our learned temperature, see Appendix Section A.1

## 3 Our Approach: AbeT

The following post-processing energy score was previously used for OOD detection in Liu et al. (2020): $\mathrm{E}(x_i; L, T_{\text{scalar}}, \theta) = -T_{\text{scalar}} \log \sum_{c=1}^{C} e^{L_c(x_i; \theta, T_{\text{scalar}})}$ . This energy score was intended to be highly negative on ID input and close to $0$ on OOD inputs via high logits on ID inputs and low logits on OOD inputs.

Our first contribution is replacing the scalar temperature with a learned one:

$$\mathrm{E}(x_i; L, T_{\text{learned}}, \theta) = - \underbrace{T_{\text{learned}}(x_i)}_{\text{Forefront Temperature Constant}} \log \sum_{c=1}^{C} e^{L_c(x_i; \theta, \overbrace{T_{\text{learned}}}^{\text{Exponential Divisor Temperature}})} \tag{1}$$

By introducing this learned temperature, there become two ways to control the OOD score: by modifying the logits and by modifying the learned temperature. We note that there are two different operations that the learned temperature performs in terms of modifying the energy score. We deem these two operations the "Forefront Temperature Constant" and the "Exponential Divisor Temperature", which are underlined and overlined, respectively, in Equation 1. Our second contribution is noting that only the Exponential Divisor Temperature contributes to the OOD score being in adherence with this property of highly negative on ID inputs and close to $0$ on OOD inputs, while the Forefront Temperature Constant counteracts that property - we therefore ablate this Forefront Temperature Constant. For a detailed explanation on this rationale, see Appendix Section A.2. Our final score (including this ablation) is as follows:

$$\mathrm{AbeT}(x_i; L, T_{\text{learned}}, \theta) = -\log \sum_{c=1}^{C} e^{L_c(x_i; \theta, T_{\text{learned}})}$$

We visualize the effects of this ablation in Figure 1, using Places365 (Zhou et al., 2018) as the OOD dataset, CIFAR-100 (Krizhevsky, 2009) as the ID dataset, and a ResNet-20 (He et al., 2016a) trained with learned temperature and a cosine logit head as the model.

For the limitations and failure cases of our method, see Appendix Section A.4.

| $D_{in}^{test}$ | CIFAR-10 | | CIFAR-100 | | ImageNet-1k | |
|---|---|---|---|---|---|---|
| Method | FPR@95 ↓ | AUROC ↑ | FPR@95 ↓ | AUROC ↑ | FPR@95 ↓ | AUROC ↑ |
| MSP | $60.5 \pm 15$ | $89.5 \pm 3$ | $82.7 \pm 11$ | $71.9 \pm 7$ | $63.9 \pm 8$ | $79.2 \pm 5$ |
| ODIN | $39.1 \pm 24$ | $92.4 \pm 4$ | $73.3 \pm 31$ | $75.3 \pm 13$ | $72.9 \pm 7$ | $82.5 \pm 5$ |
| Mahalanobis | $37.1 \pm 35$ | $91.4 \pm 6$ | $63.9 \pm 16$ | $85.1 \pm 5$ | $81.6 \pm 19$ | $62.0 \pm 11$ |
| Gradient Norm | $28.3 \pm 24$ | $93.1 \pm 6$ | $56.1 \pm 38$ | $81.7 \pm 13$ | $54.7 \pm 7$ | $86.3 \pm 4$ |
| DNN | $49.0 \pm 11$ | $83.4 \pm 5$ | $66.6 \pm 13$ | $78.6 \pm 5$ | $61.9 \pm 6$ | $82.9 \pm 3$ |
| GODIN | $26.8 \pm 10$ | $94.2 \pm 2$ | $47.0 \pm 7$ | $90.7 \pm 2$ | $52.7 \pm 5$ | $83.9 \pm 4$ |
| Energy | $39.7 \pm 24$ | $92.5 \pm 4$ | $70.5 \pm 32$ | $78.0 \pm 12$ | $71.0 \pm 7$ | $82.7 \pm 5$ |
| Energy + ReAct | $39.6 \pm 15$ | $93.0 \pm 2$ | $62.8 \pm 17$ | $86.3 \pm 6$ | 31.4* | 92.9* |
| Energy + DICE | $20.8 \pm 1$ | $95.2 \pm 1$ | $49.7 \pm 1$ | $87.2 \pm 1$ | 34.7* | 90.7* |
| Energy + ASH | $20.0 \pm 21$ | $95.4 \pm 5$ | $37.6 \pm 34$ | $89.6 \pm 12$ | $16.7 \pm 13$ | $96.5 \pm 2$ |
| LINe | 14.71** | 96.99** | 35.67** | 88.67** | 20.7* | 95.03* |
| AbeT | $\mathbf{12.5 \pm 2}$ | $\mathbf{97.8 \pm 1}$ | $\mathbf{31.1 \pm 12}$ | $\mathbf{94.0 \pm 1}$ | $40.0 \pm 11$ | $91.8 \pm 3$ |
| AbeT + ReAct | $\mathbf{12.2 \pm 1}$ | $\mathbf{97.8 \pm 1}$ | $\mathbf{26.2 \pm 7}$ | $\mathbf{94.1 \pm 2}$ | $38.1 \pm 11$ | $92.2 \pm 3$ |
| AbeT + DICE | $\mathbf{11.6 \pm 2}$ | $\mathbf{97.9 \pm 1}$ | $\mathbf{31.3 \pm 13}$ | $\mathbf{94.3 \pm 2}$ | $30.7 \pm 15$ | $93.2 \pm 3$ |
| AbeT + ASH | $\mathbf{10.9 \pm 5}$ | $\mathbf{97.9 \pm 1}$ | $\mathbf{30.6 \pm 12}$ | $\mathbf{94.4 \pm 2}$ | $\mathbf{3.7 \pm 3}$ | $\mathbf{99.0 \pm 1}$ |

\* Results where a ResNet-50 is used as in their corresponding papers instead of ResNet-101 as in our experiments. This is due to our inability to reproduce their results with ResNet-101

\*\* Results where a DenseNet is used as in their corresponding papers instead of ResNet-20 as in our experiments.

Table 1: **Comparison with other competitive OOD detection methods in classification.** OOD detection results on a suite of standard datasets compared against competitive methods which are trained with ID data only and require only one stage of training. All results are averaged across 4 OOD datasets, with the standard deviations calculated across these same 4 OOD datasets. ↑ means higher is better and ↓ means lower is better.

# 4 Experiments

In Appendix Section D, we provide intuition-building evidence to suggest that our method is able to detect OOD samples effectively (as will be shown below) despite not being exposed to explicit OOD samples at training time as a result of being exposed to misclassified ID examples in low densities of the training data during training - and then treating OOD samples at test time as akin to misclassified ID examples in low-density regions of the training data.

## 4.1 Classification Experiments

In Appendix Section B.1, we explain the experimental setup of our OOD evaluations in classification. This includes the datasets used, model architectures, training details, and hyperparameters - as well as our reasons for our choices of competitive methods.

In Table 1, we compare against competitive OOD methods outlined in Section B.1.3. All results are averaged across 4 OOD test datasets per ID dataset outlined in Section B.1.1. All OOD methods keep accuracy within $1\%$ of their respective baseline methods without any modifications to account for OOD. We note that our method achieves an average reduction in FPR@95 of $25.90\%$ on CIFAR-10, $26.55\%$ on CIFAR-100, and $77.84\%$ on ImageNet.

In Appendix Section C.2, we present evaluation studies in the presence of the inclusion of the Forefront Temperature Constant, Gradient Input Perturbation (Liang et al., 2017), the use of an Inner Product logit function instead of Cosine Similarity logit function, and an alternative architecture.

## 4.2 Semantic Segmentation Experiments

In Appendix Table 11, we compare against competitive OOD Detection methods in semantic segmentation that predict which pixels correspond to object classes not found in the training set (i.e. which pixels correspond to OOD objects). To implement our method, we replace the Inner Product per-pixel in the final convolutional layer with a Cosine Logit head per-pixel and a learned temperature layer per-pixel - then compute OOD scores per-pixel. Further experimental details can be found in

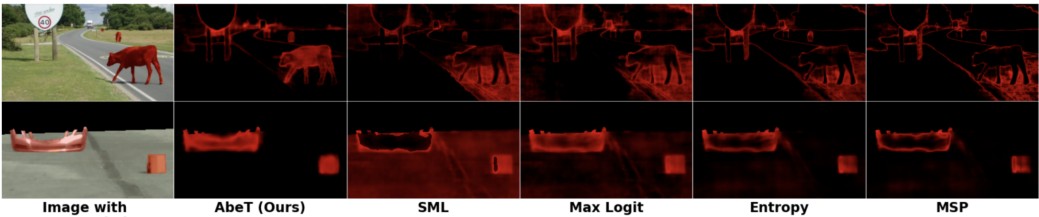

Figure 2: Qualitative comparison of OOD scores for semantic segmentation. The top row and bottom row contain examples from the datasets RoadAnomaly (Lis et al., 2019) and LostAndFound (Pinggera et al., 2016), respectively. Pixels corresponding to OOD objects are highlighted in red in each image in the leftmost column, which are cropped to regions where we have ID/OOD labels. Scores for each example (row) and technique (column) are thresholded at their respective 95% True Positive Rate and then normalized $[0, 1]$ in the red channel, with void pixels (which have no ID/OOD label) set to 0. Bright red pixels represent high OOD scores, which should cover the same region as the pixels which correspond to OOD objects in the leftmost column. We invert the scores of Standardized Max Logit, Max Logit, and MSP to allow these methods to highlight OOD pixels in red.

Appendix Section E. Notably, our method results in both a decrease in FPR@95 of $41.48\%$ and an increase in AUPRC of $34.20\%$ on average compared to previous state of the art.

We also present visualizations of pixel-wise OOD predictions for our method and methods against which we compare on a selection of images from OOD datasets in Figure 2.

### 4.3 Object Detection Experiments

In Appendix Table 12, we compare against competitive OOD Detection methods in object detection. For our experiments with `AbeT`, the learned temperature and Cosine Logit Head are directly attached to a FasterRCNN classification head's penultimate layer as described Section 2.3 - OOD scores are then computed per-box. Further experimental details can be found in Appendix Section F.2.3. We note that our method shows improved performance on ID AP [3], AUROC, and AUPRC with comparable performance on FPR@95 [4].

## 5 Conclusion

Inferences on examples far from a model's training set tend to be significantly less performant than inferences on examples close to its training set. Moreover, even if a model is calibrated on a holdout ID dataset, the confidence scores of these inferences on OOD examples are typically miscalibrated (Ovadia et al., 2019). In other words, not only does performance drop on OOD examples - users are often completely unaware of these performance drops. Therefore, detecting OOD examples in order to alert users of likely miscalibration and performance drops is one of the biggest hurdles to overcome in AI safety and reliability. Towards detecting OOD examples, we have introduced `AbeT` which mixes a learned temperature (Hsu et al., 2020) and an energy score (Liu et al., 2020) in a novel way with an effective ablation. We have established the superiority of `AbeT` in detecting OOD examples in classification, detection, and segmentation. We have additionally provided visual and empirical evidence as to why our method is able to achieve superior performance via exposure to misclassified ID examples during training time. Future work will explore if such exposure drives the performance of other OOD methods which do not train on OOD samples - as is suggested by our finding shown in Appendix Section A.4 that *all* tested OOD detection methods' failures were concentrated on misclassified ID examples. Additional follow-up work includes extending our method to the LLM setting as in LeVine et al. (2023) and the VLM setting as in LeVine et al. (2023).

---

[3]Our method shows improved ID AP via the learned temperature decreasing confidence on OOD-induced false positives.

[4]Our method provides the added benefit of being a single, lightweight modification to detectors' classification heads as opposed to significant changes to training with additional Virtual Outlier Synthesis, loss functions, and hyperparameters as in prior State of The Art (Du et al., 2022).

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

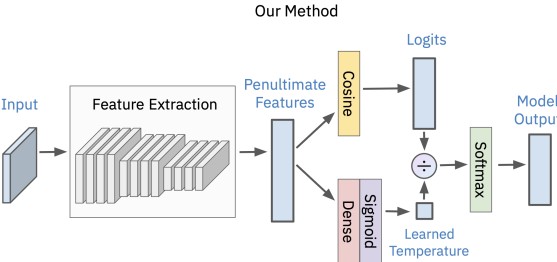

Figure 3: The network architecture of a classification network with a learned temperature.

## Appendix

# A  Our Method

## A.1  Learned Temperature Details

For all `AbeT` models, the learned temperature function is architecturally identical to that in Hsu et al. (2020): a single fully connected layer which takes inputs from the penultimate layer and outputs a single scalar per input, passes these scalars through a Batch Normalization layer, then activates these scalars via a sigmoid which normalizes them to $(0, 1)$. The learned temperature is automatically trained via the gradients induced by the back-propagation of the network's optimization since the learned temperature modifies the logits used in the forwards pass of the model. In this way, it is updated like any other layer which affects the forwards pass of the network, and thus requires no tuning, training, or datasets other than the cross-entropy-based optimization using the training dataset. The only requirement to train the learned temperature is this one-line architectural change prior to training or fine-tuning. For analysis on the insignificance of the inference time and memory costs due to the learned temperature, see Appendix Section A.5.

## A.2  Detailed Explanation For Ablating The Forefront Temperature Constant

## A.3  Detailed Explanation of Cosine-Similarity-Based Logit Function

Let $w$ and $b$ represent the weights and biases of the final layer of a network (mapping penultimate space to output space) respectively and $f^p(x_i)$ represent the penultimate representation of the network on input $x_i$. Normally, deep networks use the inner product of the $w$ and $x_i$ (plus the bias term) as the logit function: $\hat{g}_c(x_i; \theta) = w_c^T f^p(x_i) + b_c$. However, Hsu et al. (2020) found that a logit function based on the cosine similarity between $w$ and $f^p(x_i)$

$$\hat{g}_c(x_i; \theta) = \frac{w_c^T f^p(x_i)}{||w_c^T|| ||f^p(x_i)||}$$

was more effective when training logit functions that serve this dual-purpose of classifying among outputs $\{1, 2, ..., C-1, C\}$ and as an input to $S$. We therefore use this cosine-similarity score as our logit function $\hat{g}$.

## A.4  The Limitations & Failure Cases of Our Method

Because our method uses misclassified ID examples as surrogates for OOD samples (as is shown in Section D): our method does not perform well in cases where there are few misclassified ID examples during training; and most of our method's failures are on misclassified ID examples. That being said, in our experiments, *all* tested OOD detection methods' failures were concentrated on misclassified ID examples.

| Method | MSP | GODIN | Energy | Ours |
|---|---|---|---|---|
| Correctly classified vs. OOD | 36.88 | 15.58 | 16.00 | 11.02 |
| Misclassified vs. OOD | 85.23 | 42.68 | 77.13 | 38.78 |

Table 2: Comparison of a variety of OOD detection methods' FPR@95 performance at distinguishing OOD examples from correctly classified ID examples, as compared to distinguishing OOD examples from misclassified ID examples. Lower is better.

| Epoch | 1 | 3 | 6 | 7 | 8 |
|---|---|---|---|---|---|
| AUROC | 37.32 | 95.50 | 79.63 | 38.73 | 7.89 |
| Train Accuracy | 48.56 | 88.25 | 95.95 | 97.28 | 98.20 |

Table 3: Our OOD performance across epochs on MNIST. This empirically verifies our claim that our method breaks down when there becomes relatively few misclassified training samples. Higher AUROC means better OOD detection performance.

### A.4.1 The ID Failures of Our Method (And All Other Tested OOD Detection Methods) Concentrated On Misclassified ID Samples

In Table 2, we show a variety of OOD detection methods' FPR@95 performance distinguishing OOD examples from correctly classified ID examples, and distinguishing OOD examples from misclassified ID examples. We use the models and hyperparameters presented in Section B.1.2, and CIFAR10 as the ID dataset. All results are averaged across the 4 OOD datasets presented in Section B.1.1. We note that all tested OOD detection methods were significantly more capable at distinguishing OOD examples from correctly classified ID examples.

### A.4.2 Our Method In The Presence of Very Few Misclassified Training Samples

In Table 3, we present the OOD AUROC and ID accuracy metrics across training epochs, using MNIST as the ID dataset and Textures as the OOD dataset. We use the models and hyperparameters presented in Section B.1.2. As can be seen in the table, when the training accuracy greatly increases (and there becomes few misclassified training examples), our method breaks down. This shows that our method is reliant on there being a significant amount of misclassified ID training examples.

### A.5 Memory and Timing Costs Due To Learned Temperature

The time and memory costs due to the operations of the learned temperature are relatively light-weight in comparison to that of the forward pass of the model without the learned temperature. For example, with Places365 as the inference OOD dataset, the forward pass of our method took $0.94$ seconds while the forward pass of the baseline model (without the learned temperature) took $0.93$ seconds, a difference of approximately $1\%$. As an example of insignificant additional space costs necessary for the use of our method, the baseline ResNet20 (without the learned temperature) used in our CIFAR experiments has $275572$ parameters, and the learned temperature adds a mere $64$ parameters to this. This increases memory usage by less than $1\%$.

## B Experimental Setups

### B.1 Experimental Setup In Classification

### B.1.1 Classification Datasets

We follow standard practices in OOD evaluations (Huang et al., 2021; Hsu et al., 2020; Liu et al., 2020; Djurisic et al., 2022; Sun et al., 2021; Hendrycks & Gimpel, 2016; Liang et al., 2017; Sun et al., 2022; Katz-Samuels et al., 2022) in terms of metrics, ID datasets, and OOD datasets. For evaluation metrics, we use AUROC and FPR@95. We measure performance at varying number of ID classes via CIFAR-10 (Krizhevsky, 2009), CIFAR-100 (Krizhevsky, 2009), and ImageNet-1k (Huang & Li, 2021). For our CIFAR experiments, we use 4 OOD datasets standard in OOD detection: Textures (Cimpoi et al., 2014), SVHN (Netzer et al., 2011), LSUN (Crop) (Yu et al., 2015), and Places365 (Zhou et al., 2018). For our ImageNet-1k experiments we also evaluate on four standard OOD test

datasets, but subset these datasets to classes that are non-overlapping with respect to ImageNet-1k as is common practice (Huang et al., 2021; Sun et al., 2022, 2021; Sun & Li, 2021): iNaturalist (Van Horn et al., 2018), SUN (Xiao et al., 2010), Places365 (Zhou et al., 2018), and Textures (Cimpoi et al., 2014). See Appendix F.1.1 for details about these datasets.

### B.1.2 Classification Models and Hyperparameters

For all experiments with CIFAR-10 and CIFAR-100 as ID data, we use a ResNet-20 (He et al., 2016a). For all experiments with ImageNet-1k as ID data, we use a ResNetv2-101 (He et al., 2016b). We present additional experiments where we retain top OOD performance with ImageNet-1k as the ID data using an alternative architecture, DenseNet-121 (Huang et al., 2017), in Appendix Section C.2.3. All models are trained from scratch. For more experimental details, see Appendix Section F.2.1.

### B.1.3 Previous Classification Approaches

We compare against previous methods with similar constraints, training settings, and testing settings. We note that we do not compare against other methods that are trained on OOD data (Ming et al., 2022; Katz-Samuels et al., 2022; Hendrycks et al., 2018; Sharifi et al., 2024) or methods that require multiple stages of training (Khalid et al., 2022). Principally, we compare against Maximum Softmax Probability (Hendrycks & Gimpel, 2016), ODIN (Liang et al., 2017), GODIN (Hsu et al., 2020), Mahalanobis (Lee et al., 2018), Energy Score (Liu et al., 2020), Gradient Norm (Huang et al., 2021), ReAct (Sun et al., 2021), DICE (Sun & Li, 2021), LINe (Ahn et al., 2023), Deep Nearest Neighbors (DNN) (Sun et al., 2022), and ASH (Djurisic et al., 2022). For more details about these competitive methods, see Appendix Section G.

## C  Detailed Performance

### C.1  Detailed Classification Performance

In Table 1, we demonstrate the details of our superior performance in classification. All results are averaged across 4 OOD test datasets per ID dataset outlined in Section B.1.1, with the standard deviations calculated across these same 4 OOD datasets. Results from MSP (Hendrycks & Gimpel, 2016), ODIN (Liang et al., 2017), Energy (Lee et al., 2018), and Gradient Norm (Huang et al., 2021) are taken from Huang et al. (2021) and results of Mahalnobis using ImageNet-1k as the ID dataset are taken from Huang & Li (2021), as their models, datasets, and hyperparameters are identical to ours. We provide detailed results for each OOD test dataset in Appendix Section C.1.

For detailed performance of all OOD methods (baseline methods and AbeT) on each OOD dataset separately, see Table 4 for AUROC results on CIFAR, Table 5 for AUROC results on ImageNet, Table 6 for FPR@95 results on CIFAR, and Table 7 for FPR@95 results on ImageNet.

### C.2  Additional Studies In Classification

We additionally present a study of our Forefront Temperature Constant ablation in Appendix Section C.2.1 and show that this singular ablation contribution leads to a reduction in FPR@95 of $28.76\%$, $59.00\%$, and $24.81\%$ with CIFAR-10, CIFAR-100, and ImageNet as the ID datasets respectively (averaged across their 4 respective OOD datasets) compared to AbeT without the Forefront Temperature Constant ablation.

In Appendix Section C.2.4, we present Gradient Input Perturbation (Liang et al., 2017) in conjunction with our method and show that it harmed our method.

We also present experiments in Appendix Section C.2.2 where we replace the Cosine Logit Head with the standard Inner Product Head which reaffirms the finding of Hsu et al. (2020) that the Cosine Logit Head is preferable over the Inner Product Head in OOD Detection.

### C.2.1  Ablation Study

For results of AbeT without the Forefront Temperature Constant ablation, see Table 8.

| $D_{in}^{test}$ | Method | $D_{out}^{test}$ | | | | |
|---|---|---|---|---|---|---|
| | | **Textures** | **SVHN** | **LSUN** | **Places365** | **Average** |
| | MSP (Hendrycks & Gimpel, 2016) | 87.62 | 89.86 | 94.87 | 85.99 | 89.59 |
| | ODIN (Liang et al., 2017) | 89.99 | 89.63 | 99.39 | 90.81 | 92.46 |
| | Energy (Liu et al., 2020) | 88.79 | 91.97 | 99.03 | 90.55 | 92.59 |
| | Mahalanobis (Lee et al., 2018) | 92.31 | 95.99 | 97.9 | 61.15 | 86.84 |
| **CIFAR-10** | Gradient Norm (Huang et al., 2021) | 90.76 | 96.66 | 99.87 | 85.2 | 93.12 |
| | DNN (Sun et al., 2022) | 77.53 | 85.43 | 81.59 | 89.26 | 83.45 |
| | GODIN (Hsu et al., 2020) | 95.11 | 92.58 | 92.57 | 96.83 | 94.28 |
| | ReAct (Sun et al., 2021) | 91.46 | 93.03 | 92.81 | 91.92 | 92.32 |
| | DICE (Sun & Li, 2021) | 91.40 | 93.24 | 92.77 | 91.86 | 92.32 |
| | `AbeT` (Ours) | 97.17 | 97.93 | 98.09 | 98.04 | **97.81** |
| | MSP (Hendrycks & Gimpel, 2016) | 64.02 | 72.56 | 82.06 | 69.05 | 71.92 |
| | ODIN (Liang et al., 2017) | 66.35 | 66.85 | 95.09 | 72.90 | 75.30 |
| | Energy (Liu et al., 2020) | 66.74 | 76.42 | 95.90 | 72.98 | 78.01 |
| | Mahalanobis (Lee et al., 2018) | 87.48 | 81.71 | 90.74 | 50.35 | 77.57 |
| **CIFAR-100** | Gradient Norm (Huang et al., 2021) | 81.83 | 79.35 | 99.69 | 65.99 | 81.72 |
| | DNN (Sun et al., 2022) | 72.16 | 76.94 | 79.15 | 86.15 | 78.60 |
| | GODIN (Hsu et al., 2020) | 91.14 | 88.21 | 90.05 | 93.56 | 90.74 |
| | ReAct (Sun et al., 2021) | 81.11 | 87.46 | 85.46 | 83.72 | 84.44 |
| | DICE (Sun & Li, 2021) | 83.57 | 91.25 | 87.49 | 85.92 | 87.06 |
| | `AbeT` (Ours) | 95.49 | 91.55 | 93.65 | 96.44 | **94.05** |

Table 4: **AUROC comparison with other competitive OOD detection in classification methods on CIFAR.** Detection results on a suite of standard datasets compared against competitive methods which are trained with ID data only and require only one stage of training. Higher AUROC is better.

| $D_{in}^{test}$ | Method | $D_{out}^{test}$ | | | | |
|---|---|---|---|---|---|---|
| | | **Textures** | **iNaturalist** | **SUN** | **Places365** | **Average** |
| | MSP (Hendrycks & Gimpel, 2016) | 74.45 | 87.59 | 78.34 | 76.76 | 79.29 |
| | ODIN (Liang et al., 2017) | 76.30 | 89.36 | 83.92 | 80.67 | 82.56 |
| | Energy (Liu et al., 2020) | 75.79 | 88.48 | 85.32 | 81.37 | 82.74 |
| | Mahalanobis (Lee et al., 2018) | 72.10 | 46.33 | 65.20 | 64.46 | 62.02 |
| **ImageNet-1k** | Gradient Norm (Huang et al., 2021) | 81.07 | 90.33 | 89.03 | 84.82 | 86.31 |
| | DNN (Sun et al., 2022) | 78.89 | 81.61 | 88.17 | 83.27 | 82.99 |
| | GODIN (Hsu et al., 2020) | 77.67 | 86.41 | 88.51 | 83.29 | 83.97 |
| | ReAct (Sun et al., 2021) | 80.33 | 86.95 | 79.35 | 78.70 | 81.33 |
| | DICE (Sun & Li, 2021) | 83.07 | 91.59 | 86.61 | 84.67 | 86.49 |
| | `AbeT` (Ours) | 92.03 | 95.82 | 90.95 | 88.39 | **91.80** |

Table 5: **AUROC comparison with other competitive OOD detection in classification methods on ImageNet-1k.** Detection results on a suite of standard datasets compared against competitive methods which are trained with ID data only and require only one stage of training. Higher AUROC is better.

| $D_{in}^{test}$ | Method | $D_{out}^{test}$ | | | | |
|---|---|---|---|---|---|---|
| | | Textures | SVHN | LSUN | Places365 | Average |
| | MSP (Hendrycks & Gimpel, 2016) | 68.32 | 66.09 | 37.73 | 70.05 | 60.53 |
| | ODIN (Liang et al., 2017) | 52.78 | 55.52 | 2.32 | 45.86 | 39.12 |
| | Energy (Liu et al., 2020) | 58.67 | 49.80 | 3.86 | 46.48 | 39.70 |
| | Mahalanobis (Lee et al., 2018) | 28.83 | 20.91 | 9.66 | 89.24 | 37.16 |
| **CIFAR-10** | Gradient Norm (Huang et al., 2021) | 37.71 | 17.76 | 0.23 | 57.85 | 28.39 |
| | DNN (Sun et al., 2022) | 52.30 | 56.20 | 55.40 | 32.20 | 49.03 |
| | GODIN (Hsu et al., 2020) | 20.30 | 32.10 | 38.40 | 16.70 | 26.88 |
| | ReAct (Sun et al., 2021) | 52.52 | 48.09 | 48.90 | 561.75 | 50.32 |
| | DICE (Sun & Li, 2021) | 52.25 | 46.99 | 48.34 | 52.14 | 49.93 |
| | AbeT (Ours) | 15.31 | 12.37 | 10.61 | 11.74 | **12.51** |
| | MSP (Hendrycks & Gimpel, 2016) | 90.64 | 86.33 | 66.33 | 87.57 | 82.72 |
| | ODIN (Liang et al., 2017) | 89.91 | 94.80 | 26.14 | 82.57 | 73.36 |
| | Energy (Liu et al., 2020) | 88.81 | 89.03 | 21.90 | 82.55 | 70.57 |
| | Mahalanobis (Lee et al., 2018) | 42.71 | 81.46 | 68.97 | 96.50 | 72.41 |
| **CIFAR-100** | Gradient Norm (Huang et al., 2021) | 57.75 | 76.77 | 1.12 | 88.74 | 56.10 |
| | DNN (Sun et al., 2022) | 71.70 | 79.80 | 67.90 | 47.20 | 66.65 |
| | GODIN (Hsu et al., 2020) | 39.54 | 60.99 | 50.35 | 37.37 | 47.06 |
| | ReAct (Sun et al., 2021) | 71.56 | 59.95 | 63.09 | 67.82 | 65.61 |
| | DICE (Sun & Li, 2021) | 58.38 | 39.28 | 55.70 | 54.24 | 51.90 |
| | AbeT (Ours) | 21.72 | 46.85 | 35.33 | 20.86 | **31.19** |

Table 6: **FPR@95 comparison with other competitive OOD detection in classification methods on CIFAR.** Detection results on a suite of standard datasets compared against competitive methods which are trained with ID data only and require only one stage of training. Lower FPR@95 is better.

| $D_{in}^{test}$ | Method | $D_{out}^{test}$ | | | | |
|---|---|---|---|---|---|---|
| | | Textures | iNaturalist | SUN | Places365 | Average |
| | MSP (Hendrycks & Gimpel, 2016) | 82.73 | 63.39 | 79.98 | 81.44 | 76.96 |
| | ODIN (Liang et al., 2017) | 81.31 | 62.69 | 71.67 | 76.27 | 72.99 |
| | Energy (Liu et al., 2020) | 80.87 | 64.91 | 65.33 | 73.02 | 71.03 |
| | Mahalanobis (Lee et al., 2018) | 52.23 | 96.34 | 88.43 | 89.75 | 81.69 |
| **ImageNet-1k** | Gradient Norm (Huang et al., 2021) | 61.42 | 50.03 | 46.48 | 60.86 | 54.70 |
| | DNN (Sun et al., 2022) | 64.36 | 65.85 | 52.12 | 65.27 | 61.90 |
| | GODIN (Hsu et al., 2020) | 54.04 | 56.59 | 44.33 | 56.18 | 52.79 |
| | ReAct (Sun et al., 2021) | 66.70 | 57.53 | 74.20 | 76.03 | 68.62 |
| | DICE (Sun & Li, 2021) | 63.56 | 43.53 | 51.09 | 57.97 | 54.04 |
| | AbeT (Ours) | 37.82 | 25.22 | 44.71 | 52.24 | **40.00** |

Table 7: **FPR@95 comparison with other competitive OOD detection in classification methods on ImageNet-1k.** Detection results on a suite of standard datasets compared against competitive methods which are trained with ID data only and require only one stage of training. Lower FPR@95 is better.

| $D_{in}^{test}$ | Method | FPR@95 ↓ | AUROC ↑ |
|---|---|---|---|
| CIFAR-10 | AbeT | 12.51 ± 2.01 | 97.81 ± 1.43 |
| CIFAR-10 | AbeT w/o Ablation | 17.56 ± 1.69 | 96.33 ± 1.83 |
| CIFAR-100 | AbeT | 31.19 ± 12.37 | 94.05 ± 1.88 |
| CIFAR-100 | AbeT w/o Ablation | 76.08 ± 4.95 | 82.10 ± 2.26 |
| ImageNet-1k | AbeT | 43.42 ± 10.42 | 91.89 ± 1.77 |
| ImageNet-1k | AbeT w/o Ablation | 57.75 ± 15.39 | 88.58 ± 4.51 |

Table 8: **Our method with and without the ablation of the Forefront Temperature Constant.** All results are averaged across 4 OOD datasets, with the standard deviations calculated across these same 4 OOD datasets. ↑ means higher is better and ↓ means lower is better.

| $D_{in}^{test}$ | $D_{out}^{test}$ | Logit Head | FPR@95 ↓ | AUROC ↑ |
|---|---|---|---|---|
| CIFAR-10 (Cimpoi et al., 2014) | Textures | AbeT w/ Inner Product | 42.60 | 91.97 |
|  |  | AbeT w/ Cosine | 15.31 | 97.17 |
| CIFAR-10 (Netzer et al., 2011) | SVHN | AbeT w/ Inner Product | 34.00 | 93.83 |
|  |  | AbeT w/ Cosine | 12.37 | 97.93 |
| CIFAR-10 (Yu et al., 2015) | LSUN C | AbeT w/ Inner Product | 20.80 | 96.61 |
|  |  | AbeT w/ Cosine | 10.61 | 98.09 |
| CIFAR-10 (Zhou et al., 2018) | Places365 | AbeT w/ Inner Product | 24.30 | 95.30 |
|  |  | AbeT w/ Cosine | 11.74 | 98.04 |

Table 9: **Comparing our method with the Inner Product and Cosine Logit Heads.** ↑ means higher is better and ↓ means lower is better.

### C.2.2  `AbeT` With Inner Product Logit

In table 9, we compare results with the Inner Product Head as our logit function with the Cosine Logit Head as our logit function, showing the superior performance of our method with the Cosine Logit Head as is consistent with Hsu et al. (2020). These logit function definitions can be found in Section 2.3.

### C.2.3  Alternate Architecture

We also show OOD performance for DenseNet-121 (Huang et al., 2017) on ImageNet-1k in Table 10. This network was trained similarly to the ResNetv2-101, to achieve a top-1 accuracy of 72.51% on the ImageNet-1k test set. Our method maintains top OOD performance.

### C.2.4  `AbeT` With Input Perturbation

Some previous OOD techniques, like ODIN (Liang et al., 2017) and GODIN (Hsu et al., 2020), improved OOD detection with input perturbations (using the normalized sign of the gradient from the prediction). We tried applying this to our method and found that this hurt performance. Specifically, we evaluated using a ResNetv2-101 on ImageNet-1k (Krizhevsky, 2009). Across the four OOD datasets, average FPR@95 was 41.83 (an increase of 4.58%), and AUROC was 91.34 (a decrease of 0.5%) when compared to AbeT without any input perturbation. The perturbation magnitude hyperparameter was chosen using a grid search based off of GODIN (Hsu et al., 2020). Figure 4 shows performance across different perturbation magnitudes for each different OOD dataset.

## D  Understanding `AbeT`

Next, we provide intuition-building evidence to suggest that the superior OOD performance of AbeT despite not being exposed to explicit OOD samples at training time is due to exposure to misclassified ID examples during training. Towards building intuition we provide visual evidence in Figure 5

| $D_{in}^{test}$ | Method | FPR@95 ↓ | AUROC ↑ |
|---|---|---|---|
| | MSP (Hendrycks & Gimpel, 2016) | 64.47 ± 10.68 | 82.11 ± 4.77 |
| | ODIN (Liang et al., 2017) | 53.18 ± 11.04 | 87.04 ± 4.25 |
| | Energy (Liu et al., 2020) | 51.29 ± 10.28 | 87.30 ± 4.10 |
| | Mahalanobis (Lee et al., 2018) | 89.18 ± 16.94 | 46.80 ± 7.01 |
| **ImageNet-1k** | | | |
| (Krizhevsky, 2009) | Gradient Norm (Huang et al., 2021) | 41.00 ± 12.51 | 88.30 ± 4.17 |
| | DNN (Sun et al., 2022) | 68.85 ± 8.98 | 73.90 ± 9.27 |
| | GODIN (Hsu et al., 2020) | 53.23 ± 6.90 | 86.00 ± 2.92 |
| | ReAct (Sun et al., 2021) | 67.51 ± 7.28 | 81.30 ± 4.12 |
| | DICE (Sun & Li, 2021) | 68.14 ± 6.68 | 80.98 ± 4.22 |
| | AbeT (Ours) | **32.99 ± 12.54** | **92.85 ± 3.30** |

Table 10: **Comparison with other competitive OOD detection methods on large-scale datasets using an alternative architecture.** Results are on ImageNet-1k compared using a DenseNet-121 (Huang et al., 2017) and compared against competitive methods which are trained with ID data only and require only one stage of training. All results are averaged across 4 OOD datasets, with the standard deviations calculated across these same 4 OOD datasets. ↑ means higher is better and ↓ means lower is better.

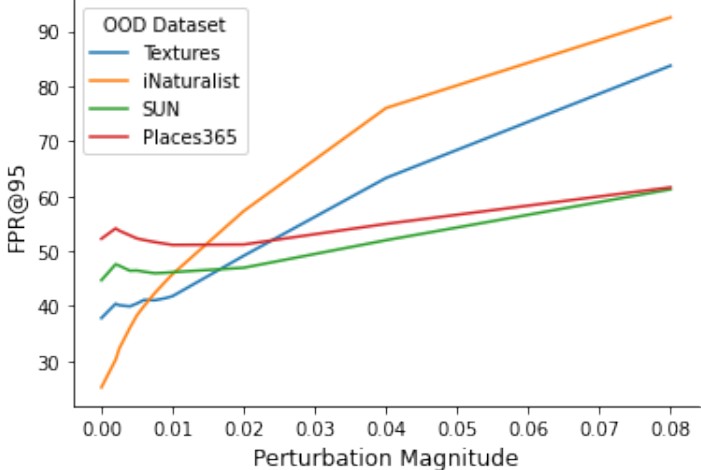

Figure 4: **Performance of AbeT with input perturbation**. This shows our method using input perturbations from ODIN (Liang et al., 2017). The x-axis is different perturbation magnitudes, and the y-axis is FPR@95 (lower is better). A ResNetv2-101 was trained on ImageNet-1k (Krizhevsky, 2009). For three of the OOD datasets, adding any perturbation hurts performance. For Places365, adding in low levels of perturbation slightly improves performance.

based on TSNE-reduced (Van der Maaten & Hinton, 2008) embeddings to support the following two hypotheses:

1. Our method learns a representation function such that OOD points are closer to misclassified ID points than correctly classified ID points, in general.

2. Our OOD scores are comparatively higher (closer to 0) on misclassified ID examples.

These combined hypotheses suggest that our score responds to OOD points similarly to the way it responds on misclassified ID points. Our scores are therefore close to 0 on OOD points due to the OOD points learning this inflation of our score (towards 0) from the (sparse) misclassified points near them, while this inflation of our score doesn't apply as much to ID points overall - as desired. This is learned without ever being exposed to OOD points at training time.

In Appendix Section D.1, we present empirical evidence to support these two hypothesis which does not utilize dimensionality reduction.

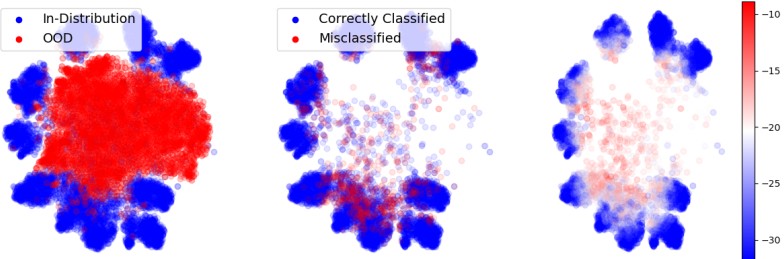

Figure 5: (**Left**) Scatter plot of OOD LSUN examples (red) and ID test CIFAR-10 examples (blue). (**Center**) ID test CIFAR-10 examples correctly classified (blue) and incorrectly classified (red). (**Right**) ID test CIFAR-10 examples colored by their `AbeT` score. Red is estimated to be more OOD. The learned temperature increasing on misclassified points (in order to deflate softmax confidence when incorrect) leads our score to inflate towards 0 on misclassified points, as can be seen in the center plot. The presence of a comparatively higher proportion of points in the center of penultimate representation space which are misclassified therefore leads to the relationship that our our score inflates towards 0 as distance to the center decreases (as can be seen on ID points in the right plot). In combination with OOD points lying in the center of penultimate representation space (as can be seen on the left plot), this means that our scores are close to 0 on OOD points - thus providing intuition (but not proof) as to why our method is able to achieve superior OOD detection performance.

### D.1 Empirical Evidence

For the following experiments, our embeddings are based on the penultimate representations of a ResNet-20 (He et al., 2016a) trained with learned temperature and a Cosine Logit on CIFAR-10 (Krizhevsky, 2009). Note that these representations are not TSNE-reduced (Van der Maaten & Hinton, 2008), since we do not aim to visualize them:

1. To empirically support that the proportion of points which are near OOD and misclassified is higher than the proportion of points which are misclassified overall, we took the single nearest neighbor in the entire ID test set for each of the OOD Places365 (Zhou et al., 2018) points in embedding space, and found that the ID accuracy on this set of OOD-proximal ID test points was $76.42\%$ compared to $91.89\%$ on all points.

2. To empirically support that misclassified ID test points have OOD scores closer to 0 than correctly classified ID test points, we found that the $99\%$ confidence intervals of the OOD scores on misclassified and correctly classified ID CIFAR-10 (Krizhevsky, 2009) are as follows, respectively: $-20.88 \pm 0.57$ and $-33.29 \pm 0.93$.

## E  Applications in Semantic Segmentation & Object Detection

### E.1 Semantic Segmentation

In Table 11, we compare against competitive OOD Detection methods in semantic segmentation that predict which pixels correspond to object classes not found in the training set (i.e. which pixels correspond to OOD objects). For evaluations, we report ID metric mIOU and OOD metrics FPR@95, AUPRC, and AUROC. For our experiments with `AbeT`, we replace the Inner Product per-pixel in the final convolutional layer with a Cosine Logit head per-pixel and a learned temperature layer per-pixel. Further details about model training can be found in Appendix Section F.2.2. We compare against methodologies which follow similar training, inference, and dataset paradigms. Notably, similar to classification, we do not compare with methods which fine-tune or train on OOD (or OOD proxy data like COCO (Lin et al., 2014b)) (Chan et al., 2021; Tian et al., 2022) or with methods which significantly change training (Mukhoti & Gal, 2018). We additionally do not compare with methods which involve multiple stages of OOD score refinement by leveraging the geometric location of the scores (Jung et al., 2021; Chan et al., 2021), as these refinement processes could be performed

| $D_{in}^{test}$ | | LostAndFound | | | RoadAnomaly | | |
|---|---|---|---|---|---|---|---|
| Method | ID mIOU | FPR@95 ↓ | AUPRC ↑ | AUROC ↑ | FPR@95 ↓ | AUPRC ↑ | AUROC ↑ |
| Entropy | 81.39 | 35.47 | 46.01 | 93.42 | 65.26 | 16.89 | 68.24 |
| MSP | 81.39 | 31.80 | 27.49 | 91.98 | 56.28 | 15.24 | 65.96 |
| SML | 81.39 | 44.48 | 25.89 | 88.05 | 70.70 | 17.52 | 75.16 |
| ML | 81.39 | 15.56 | 65.45 | 94.52 | 70.48 | 18.98 | 72.78 |
| MHLBS | 81.39 | 27 | 48 | - | 81.09 | 14.37 | 62.85 |
| AbeT | 80.56 | **3.42** | **68.35** | **99.09** | **53.5** | **31.12** | **81.55** |
| PEBAL | 80 | 0.81 | 78.29 | 99.76 | 44.58 | 45.10 | 87.63 |

Table 11: **Comparison with other competitive OOD detection methods in semantic segmentation.** OOD detection results on a suite of standard datasets compared against competitive methods which are trained with similar constraints. We present PEBAL (Tian et al., 2022) which trains on OOD data to provide context as to current performance of methods which **do** have access to OOD samples at training time, rather than presenting it as a competitive method against which we directly compare. Cityscapes is used as the ID dataset. ↑ means higher is better and ↓ means lower is better.

| Method | ID AP ↑ | FPR@95 ↓ | AUROC ↑ | AUPRC ↑ |
|---|---|---|---|---|
| Baseline | 40.2 | 91.47 | 60.65 | 88.69 |
| VOS (Du et al., 2022) | 40.5 | **88.67** | 60.46 | 88.49 |
| AbeT (Ours) | **41.2** | 88.81 | **65.34** | **91.76** |

Table 12: **Comparison with other competitive OOD detection methods in object detection.** ID model performance and OOD performance of baseline model, state of the art OOD object detection method Virtual Outlier Synthesis (Du et al., 2022), and our method, all of which do not have access to OOD at training time. ↑ means higher is better and ↓ means lower is better.

on top of any given OOD score in semantic segmentation. We compare with Standardized Max Logit (SML) (Jung et al., 2021), Max Logit (ML) (Hendrycks et al., 2022), and PEBAL (Tian et al., 2022), with results taken from Tian et al. (2022). We also compare with Mahalanobis (MHLBS) (Lee et al., 2018), with results for LostAndFound and RoadAnomaly taken from (Chan et al., 2021) and (Tian et al., 2022), respectively. Additionally, we compare to entropy (Chan et al., 2021) and Max Softmax Probability (MSP) (Hendrycks & Gimpel, 2016). We use Mapillary (Neuhold et al., 2017) and Cityscapes (Cordts et al., 2016) as the ID datasets and LostAndFound (Pinggera et al., 2016) and RoadAnomaly (Lis et al., 2019) as the OOD datasets. Details about these OOD Datasets can be found in Appendix Section F.1.2.

Notably, our method reduces FPR@95 by 78.02% on LostAndFound and increases AUPRC by 63.96% on RoadAnomaly compared to competitive methods.

We also present visualizations of pixel-wise OOD predictions for our method and methods against which we compare on a selection of images from OOD datasets in Figure 2.

### E.2 Object Detection

In Table 12, we compare against competitive OOD Detection methods in object detection. For evaluations we use ID metric AP and OOD metrics FPR@95, AUROC, and AUPRC. These evaluation metrics are calculated without thresholding detections based on any of the ID or OOD scores. For our experiments with AbeT, the learned temperature and Cosine Logit Head are directly attached to a FasterRCNN classification head's penultimate layer as described in the above sections. Further training details can be found in Appendix Section F.2.3. Our method is compared with a baseline FasterRCNN model and a FasterRCNN model using the state of the art VOS method proposed by Du et al. (2022)[5]. For datasetss we use PASCAL VOC dataset (Everingham et al., 2010) as the ID dataset and COCO (Lin et al., 2014a) as the OOD dataset.

We note that our method shows improved performance on ID AP (via the learned temperature decreasing confidence on OOD-induced false positives), AUROC, and AUPRC with comparable performance on FPR@95. Our method provides the added benefit of being a single, lightweight

---

[5]When computing OOD metrics on VOS, we use the post-processing Energy Score (Liu et al., 2020) as the OOD score, as in their paper.

modification to detectors' classification heads as opposed to significant changes to training with additional Virtual Outlier Synthesis, loss functions, and hyperparameters as in Du et al. (2022).

# F    Experimental Details

## F.1    Datasets

### F.1.1    Classification Datasets

The following information is partially taken directly from Huang et al. (2021), as the details of the datasets used in our experiments are identical to the details of the datasets used in their experiments:

**Large-scale evaluation**    We use ImageNet-1k (Huang & Li, 2021) as the ID dataset, and evaluate on four OOD test datasets following the setup in (Huang et al., 2021):

- **iNaturalist** (Van Horn et al., 2018) contains 859,000 plant and animal images across over 5,000 different species. Each image is resized to have a max dimension of 800 pixels. We evaluate on 10,000 images randomly sampled from 110 classes that are disjoint from ImageNet-1k: *Coprosma lucida, Cucurbita foetidissima, Mitella diphylla, Selaginella bigelovii, Toxicodendron vernix, Rumex obtusifolius, Ceratophyllum demersum, Streptopus amplexifolius, Portulaca oleracea, Cynodon dactylon, Agave lechuguilla, Pennantia corymbosa, Sapindus saponaria, Prunus serotina, Chondracanthus exasperatus, Sambucus racemosa, Polypodium vulgare, Rhus integrifolia, Woodwardia areolata, Epifagus virginiana, Rubus idaeus, Croton setiger, Mammillaria dioica, Opuntia littoralis, Cercis canadensis, Psidium guajava, Asclepias exaltata, Linaria purpurea, Ferocactus wislizeni, Briza minor, Arbutus menziesii, Corylus americana, Pleopeltis polypodioides, Myoporum laetum, Persea americana, Avena fatua, Blechnum discolor, Physocarpus capitatus, Ungnadia speciosa, Cercocarpus betuloides, Arisaema dracontium, Juniperus californica, Euphorbia prostrata, Leptopteris hymenophylloides, Arum italicum, Raphanus sativus, Myrsine australis, Lupinus stiversii, Pinus echinata, Geum macrophyllum, Ripogonum scandens, Echinocereus triglochidiatus, Cupressus macrocarpa, Ulmus crassifolia, Phormium tenax, Aptenia cordifolia, Osmunda claytoniana, Datura wrightii, Solanum rostratum, Viola adunca, Toxicodendron diversilobum, Viola sororia, Uropappus lindleyi, Veronica chamaedrys, Adenocaulon bicolor, Clintonia uniflora, Cirsium scariosum, Arum maculatum, Taraxacum officinale officinale, Orthilia secunda, Eryngium yuccifolium, Diodia virginiana, Cuscuta gronovii, Sisyrinchium montanum, Lotus corniculatus, Lamium purpureum, Ranunculus repens, Hirschfeldia incana, Phlox divaricata laphamii, Lilium martagon, Clarkia purpurea, Hibiscus moscheutos, Polanisia dodecandra, Fallugia paradoxa, Oenothera rosea, Proboscidea louisianica, Packera glabella, Impatiens parviflora, Glaucium flavum, Cirsium andersonii, Heliopsis helianthoides, Hesperis matronalis, Callirhoe pedata, Crocosmia × crocosmiiflora, Calochortus albus, Nuttallanthus canadensis, Argemone albiflora, Eriogonum fasciculatum, Pyrrhopappus pauciflorus, Zantedeschia aethiopica, Melilotus officinalis, Peritoma arborea, Sisyrinchium bellum, Lobelia siphilitica, Sorghastrum nutans, Typha domingensis, Rubus laciniatus, Dichelostemma congestum, Chimaphila maculata, Echinocactus texensis*

- **SUN** (Xiao et al., 2010) contains over 130,000 images of scenes spanning 397 categories. SUN and ImageNet-1k have overlapping categories. We evaluate on 10,000 images randomly sampled from 50 classes that are disjoint from ImageNet labels: *badlands, bamboo forest, bayou, botanical garden, canal (natural), canal (urban), catacomb, cavern (indoor), cornfield, creek, crevasse, desert (sand), desert (vegetation), field (cultivated), field (wild), fishpond, forest (broadleaf), forest (needle leaf), forest path, forest road, hayfield, ice floe, ice shelf, iceberg, islet, marsh, ocean, orchard, pond, rainforest, rice paddy, river, rock arch, sky, snowfield, swamp, tree farm, trench, vineyard, waterfall (block), waterfall (fan), waterfall (plunge), wave, wheat field, herb garden, putting green, ski slope, topiary garden, vegetable garden, formal garden*

- **Places365** (Zhou et al., 2018) is another scene dataset with similar concept coverage as SUN. A chosen subset of 10,000 images across 50 classes (not contained in ImageNet-1k) are used: *badlands, bamboo forest, canal (natural), canal (urban), cornfield, creek, crevasse, desert (sand), desert (vegetation), desert road, field (cultivated), field (wild), field road,*

*forest (broadleaf), forest path, forest road, formal garden, glacier, grotto, hayfield, ice floe, ice shelf, iceberg, igloo, islet, japanese garden, lagoon, lawn, marsh, ocean, orchard, pond, rainforest, rice paddy, river, rock arch, ski slope, sky, snowfield, swamp, swimming hole, topiary garden, tree farm, trench, tundra, underwater (ocean deep), vegetable garden, waterfall, wave, wheat field*

- **Textures** (Cimpoi et al., 2014) contains 5,640 real-world texture images under 47 categories. We use the entire dataset for evaluation.

**CIFAR benchmark**     CIFAR-10 and CIFAR-100 (Krizhevsky, 2009) are widely used as ID datasets in the literature, which contain 10 and 100 classes, respectively. We use the standard split with 50,000 training images and 10,000 test images. We evaluate our approach on four common OOD datasets, which are listed below:

- **SVHN** (Netzer et al., 2011) contains color images of house numbers. There are ten classes of digits 0-9. We use the entire test set containing 26,032 images.
- **LSUN C** (Yu et al., 2015) contains 10,000 testing images across 10 different scenes. Image patches of size 32×32 are randomly cropped from this dataset.
- **Places365** (Zhou et al., 2018) contains large-scale photographs of scenes with 365 scene categories. There are 900 images per category in the test set. We randomly sample 10,000 images from the test set for evaluation.
- **Textures** (Cimpoi et al., 2014) contains 5,640 real-world texture images under 47 categories. We use the entire dataset for evaluation. test set for evaluation.

### F.1.2   Semantic Segmentation Datasets

For semantic segmentation experiments, we treat Mapillary (Neuhold et al., 2017) and Cityscapes (Cordts et al., 2016) as the ID datasets and LostAndFound (Pinggera et al., 2016) and RoadAnomaly (Lis et al., 2019) as the OOD datasets.

- **Mapillary Vistas** (Neuhold et al., 2017) is an urban street scenes dataset consisting of 25,000 images with labels spanning 124 categories intended for autonomous driving. The dataset provides examples covering 6 continents and a variety of weathers and seasons.
- **Cityscapes** (Cordts et al., 2016) is an urban street scenes dataset consisting of 5,000 images with fine annotations and an additional 20,000 images with coarse annotations. The dataset covers 30 semantic classes and covers 50 cities over several seasons.
- **LostAndFound** (Pinggera et al., 2016) is an urban street scene dataset comprising of 1203 real images which contain roads with anomalous objects like cones, boxes, tires, or toys. The dataset spans 13 different scenes and features 37 different types of anomalous objects and provides labels for the road, anomalous objects, and background.
- **RoadAnomaly** (Lis et al., 2019) is a rural street scene dataset comprised of 60 web-scraped images of rural roads with obstacles like zebras, cows, or sheep. The varied scale and size of the anomalous objects, in addition to the rural background, make this dataset very difficult for OOD detection methods.

### F.1.3   Object Detection Datasets

- **PASCAL VOC** (Everingham et al., 2010) is a natural image object detection dataset that contains 20 different object classes. The dataset contains 2,913 distinct images.
- **COCO** (Lin et al., 2014a) is a large scale object detection dataset. It contains 91 different object categories with over 200,000 labelled images. Image resolution of this dataset is 640 x 480.

## F.2   Models and Hyperparameters

### F.2.1   Classification Models and Hyperparameters

For all CIFAR experiments, we trained with a batch size of 64 and - identical to Huang et al. (2021) - with an initial learning rate of 0.1 which decays by a factor of 10 at epochs $50\%, 75\%$, and $90\%$

of total epochs. For all Imagenet experiments, we trained with a batch size of 512 with an initial learning rate of 0.1 which decays by a factor of 10 at epochs 20 and 30. For our CIFAR and ImageNet experiments we train for 200 and 40 total epochs respectively without early stopping or validation saving. For CIFAR experiments, we use Horizontal Flipping and Trivial Augment Wide (Müller & Hutter, 2021). For Imagenet experiments, we use the augmentations from Torchvision's recipe (TODO: cite blog https://pytorch.org/blog/how-to-train-state-of-the-art-models-using-torchvision-latest-primitives/).

### F.2.2 Semantic Segmentation Models and Hyperparameters

For all semantic segmentation experiements, we utilize the DeepLabv3+ segmentation model with a WideResnet38 backbone (Zhu et al., 2019; Reda et al., 2018). All models are pretrained on Mapillary (Neuhold et al., 2017) and finetuned on Cityscapes (Cordts et al., 2016). For comparison methods, we use the we use the publicly available weights (Reda et al., 2018) as the pretrained model. For our method, in order to incorporate our Cosine Logit head and learned temperature layer, we trained a new model from scratch following the exact training in (Zhu et al., 2019; Reda et al., 2018). This consists of a pretraining stage on Mapillary, which trains for a maximum of 175 epochs with an initial learning rate of 0.02, decaying each epoch by a factor of $(1 - \frac{epoch}{max\_epoch})$. Additionally, several advanced training techniques are used, such as class uniform sampling, augmentations like random cropping and Gaussian blur, Synchronized Batch Normalization, and a special inverse-target-frequency weighted cross-entropy loss. Once the Mapillary pretrained model reaches a mIOU of $\geq 0.5$, the Cityscapes finetuning training begins. The Cityscapes finetuning consists of training for a maximum of 175 epochs with an initial learning rate of 0.001, which begins to decay by a factor of $1 - \frac{epoch}{max\_epoch}$ each epoch until epoch 100, at which point the learning rate decays by $(1 - \frac{epoch-100}{max\_epoch-100})^{1.5}$ per epoch. The same advanced training techniques as above are utilized, except with a custom loss function designed to reach greater accuracy on borders between classes. The finetuning is considered finished when the model reached a Cityscapes Validation mIOU $\geq 0.8$. We do not utilize their optional additional training step using label propogation via video prediction and the Cityscape sequences dataset due to compute constraints. For more specifics about training, visit https://github.com/NVIDIA/semantic-segmentation/tree/sdcnet (TODO: hyperlink). For both the Mapillary pretraining and the Cityscapes finetuning, we utilize a batch size of 8 on 8 NVIDIA T4 GPUs and conduct all training in half-precision for speed.

For evaluation, we utilize the code and framework from Chan et al. (2021), which uses a histogram approach to avoid memory errors while calculating statistics over thousands of images. OOD scores are normalized to a range of $[0, 1]$, then bucketed into either the ID histogram or OOD histogram of 100 even spaced bins spanning $[0, 1]$ depending on their label. For example, entropy scores are normalized to $[0, 1]$ by dividing the entropy scores by the logarithm of the number of classes. At this point, we invert some scoring methods like MSP (using an inversion that exactly reverses the ordering of points according to OOD score) so that all methods align with the framework of ID scores near 0 and OOD scores near 1. Once all the scores have been added to their respective histograms, the counts are normalized by a maximum value, chosen as $10^7$, then transformed back into lists of predictions according to their bin value. Statistics like FPR@95, AUROC, and AUPRC are calculated on this list of scores, which drastically drops the runtime while maintaining the correct distribution. For further details on evaluation techniques, see Chan et al. (2021).

### F.2.3 Object Detection Models and Hyperparameters

For all object detection experiments, we utilize the detectron2 package in order to train a FasterRCNN with a ResNet50 backbone pretrained on ImageNet. All models were trained with a batch size of 4. During training, images were augmented with random croppinxg and flipping. A base learning rate of 0.02 was used and decays at steps 12,000 and 16,000 per the setup in (Du et al., 2022). All models were trained on an Nvidia V100 GPU for a maximum of 18,000 iterations, where only the model with the best validation loss was saved.

# G   Classification Baseline Methods

## G.1   Hyperparameters

For all previous approaches, we use the default hyperparameters described in their respective papers, other than for our CIFAR-10 experiment we use $K = 200$ for Deep Nearest Neighbors (Sun et al., 2022), as we found it to reduce FPR@95 by $12.40\%$ as compared to the default $K = 50$ in the paper. For our ImageNet experiments for Deep Nearest Neighbors (Sun et al., 2022), we use a sampling ratio of $1\%$ for runtime memory reasons.

## G.2   Descriptions

For the reader's convenience, we summarize in detail a few common techniques for defining OOD scores in classification that measure the degree of ID-ness on the given sample. Some of these descriptions are taken directly from Huang et al. (2021).

The following scores follow a convention that a higher (resp. lower) score is indicative of being ID (resp. OOD):

**MSP** Hendrycks & Gimpel (2016) propose to use the maximum softmax score to detect OOD samples.

**ODIN** Liang et al. (2017) improved OOD detection with temperature scaling and input perturbation. Note that this is different from calibration, where a much milder T will be employed. While calibration focuses on representing the true correctness likelihood of ID data, the OOD scores proposed by ODIN are designed to maximize the gap between ID and OOD data and may no longer be meaningful from a predictive confidence standpoint.

**GODIN** Hsu et al. (2020) substitutes the temperature scaling with the use of a explicit variable $d_{in}$ in the classifier, rewriting the class posterior probability as the quotient of the joint class-domain probability and the domain probability using the rule of conditional probability $p(y|d_{in}, x_i) = \frac{p(y, d_{in}|x_i)}{p(d_{in}|x_i)}$. GODIN uses this dividend/divisor structure to define a logit per class as the division of two functions $f_j(x) = \frac{h_j(x)}{g_j(x)}$. The learned $h$ and $g$ from ID map well to the quotient decomposition above, and are used as the OOD score. We found using $g_j(x)$ as the OOD score to be the most performant on average, and thus we use this divisor as the GODIN OOD score for all experiments. We additionally do not perform backward-pass-based input perturbations with GODIN, as we found them to harm OOD performance - similar to the findings of Huang et al. (2021) on ODIN. This is equivalent to setting $\epsilon = 0$.

**DICE** Sun & Li (2021) proposes a sparsification-based OOD detection framework. The key idea is to rank weights in the penultimate layer based on a measure of contribution, and selectively use the most salient weights to derive the output for OOD detection. For ID data, only a subset of units contributes to the model output. In contrast, OOD data can trigger a non-negligible fraction of units with noisy signals. To exploit this, DICE ranks weights based on the measure of contribution (weight x activation), and selectively uses the most contributing weights to derive the output for OOD detection. As a result of the weight sparsification, the model's output becomes more separable between ID and OOD data. Importantly, DICE can be conveniently used by post hoc weight masking on a pre-trained network and therefore can preserve the ID classification accuracy.

**ReAct** In the penultimate layer, the mean activation for ID data is well-behaved with a near-constant mean and standard deviation. In contrast, for OOD data, the mean activation has significantly larger variations across units and is biased towards having sharp positive values (i.e., positively skewed). As a result, such high unit activation can undesirably manifest in model output, producing overconfident predictions on OOD data. The method Rectified Activations (dubbed ReAct, proposed by Sun et al. (2021)) uses the above observation for OOD detection. In particular, the outsized activation of a few selected hidden units can be attenuated by rectifying the activations at an upper limit $c > 0$. Conveniently, this can be done on a pre-trained model without any modification to training. After rectification, the output distributions for ID and OOD data become much more well-separated. Importantly, this truncation largely preserves the activation for ID data, and therefore ensures the classification accuracy on the original task is largely comparable.

**Max Logit** (Hendrycks et al., 2022) attempts to address a shortcoming of MSP where the softmax operator may redistribute probability mass among several large logits. In the case of two large logits, their relative maximum is lowered by the softmax operator as they must split the probability mass. The Max Logit approach is to simply observe the maximum logit as the OOD score instead of the maximum softmax probability.

**Standardized Max Logit** (Jung et al., 2021) improves upon the Max Logit approach by observing that the distribution of max logits for each class is significantly different from one another. The SML approach is to standardize the max logit scores on a per-class basis using the ID dataset to collect expected means and variances. These per-class means and variances are then used during OOD evaluation, with the Z-score of the respective max logit for each prediction used as the OOD score.

The following scores follow a convention that a higher (resp. lower) score is indicative of being OOD (resp. ID):

**Energy** Liu et al. (2020) first proposed using energy score for OOD uncertainty estimation. The energy function maps the logit outputs to a scalar $S_{Energy}(x_i; f) \in R$, which is relatively more negative for ID data $S_{Energy}(x_i; f) = -T \log \sum_{j=1}^{C} e^{f_j(x_i)/T}$. We compare against the version of Energy Score which is not fine-tuned on OOD data at training time, as training on OOD data would violate our assumption that we do not have access to OOD data at training time.

**Mahalanobis** Lee et al. (2018) use multivariate Gaussian distributions to model class-conditional distributions in the penultimate layer and use Mahalanobis distance-based scores to these distributions for OOD detection.

**Entropy** (Chan et al., 2021) propose using the entropy of the softmax probabilities as the scoring method for OOD detection. They propose the discrete entropy formula $E(f(x)) = -\sum_{j \in C} f_j(x) log(f_j(x))$, where $f_j$ represents the softmax probability over each class $j \in C$. This method sees strong performance when paired with OOD finetuning where the model is forced to learn to output uniform probabilities on OOD examples in order to maximize the entropy of those predictions.

**Deep Nearest Neighbors** Methods like (Lee et al., 2018) make a strong distributional assumption of the underlying feature space being class-conditional Gaussian. Sun et al. (2022) explore the efficacy of non-parametric nearest-neighbor distance for OOD detection. In particular, they use the distance to the k-th nearest neighbor in the penultimate space of the training set as their OOD score. We compare against the version of Deep Nearest Neighbors which does train a representation network via Supervised Contrastive Loss (Khosla et al., 2020). Training a representation network with Supervised Contrastive Loss would modify training to violate our assumption that we only train in a single stage, as there would be a required second training stage which fits a model that maps these representations to logit space $\mathbb{R}^C$.

**LINe** (Ahn et al., 2023) is an improvement on Sun & Li (2021), where the model's activation are first clipped. Shapley-based pruning is then run on both the activations and weights to get a final OOD score.

