# OpenReview forum: "Out-of-Distribution Detection & Applications With Ablated Learned Temperature Energy"
_NeurIPS.cc/2024/Workshop/BDU — NeurIPS BDU Workshop 2024 Poster_

### Official Review · Reviewer_injH · 2024-09-22
**Comments for paper #40**

**Rating:** 6
**Confidence:** 3

**Review:**

Summary:
This paper presents Ablated Learned Temperature Energy (AbeT), a method for detecting Out-of-Distribution (OOD) inputs in deep neural networks. AbeT reduces the False Positive Rate at 95% True Positive Rate by 43.43% compared to existing methods, without needing multiple training stages or hyperparameters. The model effectively learns to differentiate between In-Distribution (ID) and OOD samples through exposure to misclassified ID examples. Additionally, it improves OOD detection in object detection and semantic segmentation tasks, achieving significant performance gains.

Strengths:
1. The evaluation scenarios are comprehensive. The authors have conducted experiments to verify the effectiveness of proposed detection scheme across classification, object detection, and semantic segmentation, making proposed approach more convincing.
2. The writing is good. The authors express the idea properly and effectively.


Weaknesses:
1. Lack of discussions of recent works. In the introduction section, the latest work that the authors have discussed about is 2020, which is  kind of out-of-date. I suggest the authors to include more recent studies, such as [1].

[1] Fang Z, Li Y, Lu J, et al. Is out-of-distribution detection learnable? Advances in Neural Information Processing Systems, 2022, 35: 37199-37213.

---

### Official Review · Reviewer_X29s · 2024-09-26
**This paper proposes a simple method for OOD detection, but there are concerns regarding its limited novelty and the presentation.**

**Rating:** 6
**Confidence:** 4

**Review:**

Summary:
This paper proposes a method for out-of-distribution (OOD) detection by incorporating a learned temperature in the post-processing energy score, while ablating the temperature in the foreground. The experiments demonstrate the method’s effectiveness across multiple tasks, including classification, segmentation, and object detection.

Strengths:

The method is simple and does not require additional training steps or the use of OOD samples during training.
Experimental results indicate its usefulness across various applications, showing strong practical relevance.
The research addresses an important problem in OOD detection, with potential applications in several fields.

Weaknesses:

The method is primarily empirical and builds on modifying an existing post-processing energy score, which limits the novelty of the contribution.

Questions/Concerns:

Mathematical proof: The paper would benefit from a mathematical proof of the proposed method rather than relying solely on empirical visualizations.
Presentation of results: The key results are placed in the appendix/supplementary material rather than the main paper. Supplementary material should ideally provide additional or supporting information, not the main findings. This structure makes the paper feel incomplete and may also conflict with the page limit rules for the main submission.

---

### Decision · Program_Chairs · 2024-10-09

Accept (Poster)